# NN4SysBench: Characterizing Neural Network Verification for Computer Systems

**Shuyi Lin**[1]    **Haoyu He**[2,3]    **Tianhao Wei**[4]    **Kaidi Xu**[5]    **Huan Zhang**[6]
**Gagandeep Singh**[6]    **Changliu Liu**[4]    **Cheng Tan**[1]
[1]Northeastern University   [2]University of Tübingen   [3] Tübingen AI Center
[4]CMU   [5]Drexel University   [6]UIUC

## Abstract

We present NN4SysBench, a benchmark suite for neural network verification that is composed of applications from the domain of computer systems. We call these *neural networks for computer systems* or *NN4Sys*. NN4Sys is booming: there are many proposals for using neural networks in computer systems—for example, databases, OSes, and networked systems—many of which are safety-critical. Neural network verification is a technique to formally verify whether neural networks satisfy safety properties. We however observe that NN4Sys has some unique characteristics that today's verification tools overlook and have limited support. Therefore, this benchmark suite aims at bridging the gap between NN4Sys and the verification by using impactful NN4Sys applications as benchmarks to illustrate computer systems' unique challenges. We also build a compatible version of NN4SysBench, so that today's verifiers can also work on these benchmarks with approximately the same verification difficulties. The code is available here: `https://github.com/Khoury-srg/NN4SysBench`.

## 1   Introduction

AI safety is critical in the field of deep learning and neural networks. One pivotal technique to guarantee safety is *neural network verification* [23, 30], or *NN-verification*. It offers systematic and rigorous guarantees to ensure the reliability and robustness of neural networks. In recent years, formally verifying neural networks has gained significant attention, and various benchmark suites [3] have emerged to assess the verification efficiency for different problems. These benchmark suites—such as ACAS XU [22, 23], MNIST [28], and CIFAR-10 [27, 17]—provide a platform for researchers and practitioners to compare different verification methodologies, ultimately helping people use neural networks safely.

However, there has been limited attention on verifying *neural networks for computer systems* (NN4Sys). NN4Sys refers to neural networks that replace traditional components in computer systems, like OSes, databases, and networked systems. People have demonstrated that NN4Sys can significantly improve the performance of a system [26, 33]. Meanwhile, since computer systems are critical infrastructures, NN4Sys has strong safety requirements. But, unlike traditional ML tasks, NN4Sys has unique characteristics which bring new challenges for today's verifiers. (We detail the motivation and the challenges in §2.) Thus, there is a strong need to connect NN4Sys and NN-verification.

To bridge the gap between NN4Sys and NN-verification, we introduce *NN4SysBench*, a benchmark suite for NN-verification to verify impactful NN4Sys applications. NN4SysBench encompasses a diverse set of applications from a wide spectrum of systems, including database indexes [26], distributed system schedulers [35], and Internet congestion control [21]. Each benchmark has a set of specifications that cover different verification difficulties. Over the past three years, NN4SysBench has

been used as part of the benchmarks in the International Verification of Neural Networks Competition (VNN-COMP) [3], contributing to the evaluation of the state-of-the-art verifiers.

**Goals and non-goals.** As mentioned earlier, NN4Sys brings new challenges to verifiers, and until today, some NN4Sys properties still haven't been supported. Therefore, NN4SysBench has two versions—a vanilla version and a compatible version—targeting different goals:

- *Vanilla version*: this version contains the original neural networks proposed by their original authors, and the specifications that should have been verified assuming an oracle verifier.
- *Compatible version*: we customize the networks to simulate the original proposals, and remove the unsupported operators and network architectures, so that verifiers can verify these networks.

The primary goal of NN4SysBench is to demonstrate the characteristics of NN4Sys, with the overarching aim of helping and accelerating advancements in future neural network verifiers. To achieve this, the vanilla version is built to exemplify verification challenges that NN4Sys poses, thereby serving as motivations for new verification techniques. Meanwhile, the compatible version is used for evaluating the performance of today's verifiers. It provides an approximation of the verification difficulties inherent in various NN4Sys applications. Beyond its immediate usefulness, we envision that the compatible NN4SysBench will also serve as a stepping stone towards the eventual establishment of verifying vanilla NN4Sys.

However, NN4SysBench is not designed for the following points: first, NN4SysBench is not built for detecting unsafe behaviors in the original NN4Sys proposals. In our benchmarks, we replicate the original proposal by retraining the neural networks. A specification violation on our networks does not necessarily imply that the original proposal is unsafe. Second, NN4SysBench is not designed for proposing perfect specifications. We do not claim that the specifications used in our benchmarks are what NN4Sys must satisfy. The specifications are subjective to our understanding of the applications, and many specifications are "better-to-obey" rather than "have-to-obey". In fact, it is an open question to design high-quality specifications for NN4Sys. Third, NN4SysBench does not contribute to NN-verification. It does not invent any new verification methods or algorithms.

**Contributions.** Our contributions are as follows.

- We design and implement NN4SysBench, a benchmark suite that bridges NN4Sys and NN-verification. We propose safety specifications for these NN4Sys applications.
- We customize the existing NN4Sys to form a compatible version of NN4SysBench, which enables today's verifiers to experiment on NN4Sys applications.
- We highlight some NN4Sys characteristics that we observe, some of which significantly differ from classic machine learning tasks.
- We experiment NN4SysBench with two state-of-the-art verifiers, $\alpha\beta$-CROWN [55, 48] and Marabou [24]. They perform differently on different benchmarks, which indicates that NN4SysBench covers various verification difficulties.

We wish that NN4SysBench can illustrate how NN4Sys works, meanwhile also demonstrate the potentials of NN-verification for computer systems. NN4SysBench may further hint better ways for verifying NN4Sys applications.

## 2   Motivation: verifying NN4Sys

**Background: NN4Sys.** People have introduced a broad range of NN4Sys applications in computer systems. These applications either use neural networks to replace traditional data structures for performance (e.g., learned index [37]), or to make better decisions (e.g., learned scheduler [42]). Below we list a few NN4Sys applications in databases, OSes, and networked systems.

In databases, people proposed learned index [26, 13, 12, 45, 37], learned cardinality estimation [25], and learned query optimizers [38, 36]. In operating systems, there are systems using neural networks for predicting I/O latency [20], page prefetching [7], load balancing [16], and job scheduling [41]. Likewise, neural networks are used for networked systems, including congestion control [21], datacenter network traffic optimization [9, 43], resource allocation and scheduling [32, 54, 35], optimizing video streaming [33], and packet classification [29]. In addition, Haj-Ali et al. [19] and Mao et al. [34] have surveyed NN4Sys applications.

**Why verifying NN4Sys?** Despite having better average performance, NN4Sys is not widely deployed in practice. One major reason is that neural networks are black boxes and what has been learned by the neural networks is unclear to developers. Thus, neural networks may produce unexpected results for unseen inputs, and risk the stability and correctness of the systems. For example, a neural network based scheduler may attempt to schedule invalid jobs [32]; a learned video streaming system may pick the worst bitrate even if the network condition is good [15]; a learned index may output a faraway data position for non-existing keys [50]; and a learned cardinality may violate monotonicity and predict a smaller number for a larger query range [49].

Meanwhile, testing cannot address the above-mentioned problem, as the input space is usually infinite (like database key space) and testing cannot cover all possible inputs. Despite the challenge of infinite input spaces, there is a hope: neural network verification can provably check networks for a continuous range of inputs. This enables developers to verify if a trained NN4Sys follows a safety property for any inputs under some predefined conditions, even if infinite. In fact, NN-verification has already been used for examining desired properties of networked systems [15, 11].

**NN4Sys brings new verification challenges.** Though promising, adopting verification in NN4Sys is non-trivial. This is because NN4Sys has its own characteristics, some of which turn out to be challenging for today's neural network verifiers. Here, we list the unique characteristics of NN4Sys to highlight some challenges to apply NN-verification on NN4Sys applications.

1. *Large number of specification entries.* We observe that NN4Sys usually has many specification entries—a basic specification unit, representing a single rule. For example, in reachability specifications, an entry comprises an input range and an output range. This high number of entries is because safety properties need to cover the entire input space to be comprehensive. For example, the learned index in our benchmark has 150K entries.
2. *Monotonicity specification.* Beyond normal specifications of specifying input-output constraints, NN4Sys also requires monotonicity properties. As an example, learned cardinality estimation requires results to be monotonically increasing while query ranges increase.
3. *Probabilistic specification.* NN4Sys sometimes needs specifications to be probabilistic—it can be either the guarantees hold probabilistically (e.g., in probabilistic data structures) or developers want to give some leeway to networks (e.g., allowing false predictions).
4. *Temporal specification.* in many cases, NN4Sys requires properties related to time. For example, congestion control protocols should eventually increase the packet sending rate when facing good network conditions, but increasing doesn't have to happen immediately.
5. *Hierarchical models.* NN4Sys sometimes uses multiple neural networks in a hierarchical structure which together serve one task, for example, RMI [26]. Ideally, verification tools can check them end-to-end in one pass.

To clarify the challenges, here are several nuances to mention. First, we are not claiming that every challenge mentioned is impossible to verify; some are merely inefficient due to the absence of targeted support. Similarly, even if some challenges are not directly supported, there are alternative methods to verify them by making modifications to networks or specifications—as demonstrated in how we build the compatible version of NN4SysBench. It's also worth noting that the verification of certain challenges is exclusive to specific verifiers and lacks a universal verification interface. Lastly, there are indeed some challenges that are currently unverifiable (e.g., variable tensor indexing), presenting challenges to today's verification techniques.

**Bridging the gap with NN4SysBench.** To address the divide between NN4Sys and NN-verification, we propose NN4SysBench, as a benchmark suite for NN4Sys. NN4SysBench is designed to include impactful NN4Sys that already exists, plus specifications that we create. Also, each benchmark is available in two versions: vanilla and compatible. So far, NN4SysBench includes six benchmarks that span a wide array of applications. These have been implemented in diverse computer systems, each providing unique functionalities and specifications. Next, we introduce NN4SysBench.

## 3 NN4SysBench

NN4SysBench contains six NN4Sys applications, detailed in Figure 1. We will introduce these applications (§3.1), their specifications (§3.2), and the overall organization of NN4SysBench (§3.3).

| benchmark | bench size | training | network | | | operators | | specifications | | | |
|---|---|---|---|---|---|---|---|---|---|---|---|
| | | | size | depth | in_dim | compatible | vanilla | reach | mono | prob | temp |
| LearnedIndex | small | SL | 33K | 8 | 1 | Gemm,ReLU | – | ✓ | | | |
| | large | | 66K | 12 | 1 | | | | | | |
| CardEsti | small | SL | 103K | 19 | 154 | Gemm,ReLU, | | ✓ | | | |
| | large | | 25M | 18 | 154 | MatMul,Split, | | ✓ | | | |
| (dual-model) | small | SL | 103K | 21 | 308 | ReduceSum,Slice, | – | | ✓ | | |
| | large | | 25M | 20 | 308 | Concat,Sigmoid | | | ✓ | | |
| BloomFilter | – | SL | 133K | 8 | 1 | Gemm,ReLU | – | | | ✓ | |
| CongestCtrl | small | RL | 0.6K | 6 | 30 | Gemm,Tanh | – | ✓ | | | |
| | medium | | 1.5K | | | | | ✓ | | | |
| | large | | 4.1K | | | | | ✓ | | | |
| (dual-model) | small | RL | 0.6K | 8 | 60 | Gemm,Tanh,Split | – | | ✓ | | |
| | medium | | 1.5K | | | | | | ✓ | | |
| | large | | 4.1K | | | | | | ✓ | | |
| (chain-model) | small | RL | 0.6K | 40 | 151 | Gemm,Tanh | – | | | | ✓ |
| | medium | | 1.5K | | | Split,Concat | | | | | ✓ |
| | large | | 4.1K | | | | | | | | ✓ |
| LearnedSched | – | RL | 3K | 217 | 4.3K | Gemm,ReLU, Split,Reshape, MatMul(const) | +LeakyReLu, +Slice,+Gather, +MatMul(var) | ✓ | | | |
| AdaptBitrate | small | RL | 103K | 12 | 48 | Gemm, ReLU, | +Slice, +MatMul, +Conv1d | ✓ | | | |
| | medium | | 264K | 11 | | | | ✓ | | | |
| | large | | 527K | 11 | | | | ✓ | | | |
| (dual-model) | small | RL | 103K | 18 | 96 | Reshape, Concat | +Flatten, +ReduceSum, +Gather | | ✓ | | |
| | medium | | 264K | | | | | | ✓ | | |
| | large | | 527K | | | | | | ✓ | | |

Figure 1: NN4SysBench application overview. We get network parameters from onnx-tool [6]: "size" is the number of trained parameters in network; "depth" is the longest path in network's computational graph; "in_dim" is the size of flattened input tensor. In "training", "SL" means supervised learning; "RL" means reinforcement learning. In "operators", "vanilla" column indicates operators used by the original models but not by the compatible models. In "specifications", "reach", "mono", "prob", "temp" represent reachability, monotonicity, probabilistic, and temporal specifications, respectively.

## 3.1 Neural networks for computer systems

The six applications in NN4SysBench are: learned index, learned bloom filter [26], learned cardinalities [25], learned congestion control [21], learned adaptive bitrate [33], and learned scheduler [35]. Below, we introduce them by their training methods.

**Supervised learning.** There are three NN4Sys applications that use supervised learning: (1) Database learned index (abbreviated as *LearnedIndex*): A database index is a data structure that improves data retrieval by linking database keys to their storage positions. Learned indexes replace traditional structures like B-Trees with ML models that predict storage locations based on database keys. (2) Learned bloom filter (abbreviated as *BloomFilter*): a bloom filter is a probabilistic data structure that has been widely used in many computer systems. Bloom filters test whether an element (for example, a string) is in a pre-defined set. Bloom filters allow false positives: it may return true for an element that is not in the set. The inputs are being-tested elements, and the outputs are booleans, whether the elements are in the set. (3) Learned cardinalities (abbreviated as *CardEsti*): database cardinality estimation predicts the number of rows returned by a database query (i.e., a SQL statement), which will then influence the query optimization plans. Traditional cardinality estimation relies on heuristics and domain knowledge; whereas, the learned cardinalities learn from the trace data. The inputs of learned cardinalities are SQL queries, and the outputs are estimated number of returned rows.

**Reinforcement learning.** The other three use reinforcement learning: (1) Learned Internet congestion control (abbreviated as *CongestCtrl*): congestion control protocols are to ensure efficient data transmission and prevent packet congestion in a network. Learned congestion control studies the patterns of congestion and non-congestion conditions, takes network conditions as input, and decides sending rates in the near future. (2) Learned adaptive bitrate (abbreviated as *AdaptBitrate*): in video streaming, adaptive bitrate algorithms are used on client-side video players to decide the resolution (e.g., 720P) to download for the next video chunk. Learned adaptive bitrate uses neural networks to select future video chunks based on observations collected by client video players. (3) Learned distributed system scheduler (abbreviated as *LearnedSched*): a distributed computing system like Spark manages a cluster of machines and runs multiple jobs, each containing multiple

| benchmark | spec | description |
|-----------|------|-------------|
| LearnedIndex | reach | All predicted data locations are error-bounded. |
| CardEsti | reach | The predicted cardinalities are close to the ground-truth cardinalities from the database. |
| | mono | A larger-ranged query returns larger cardinality. |
| BloomFilter | prob | The false positive and false negative rates are bounded. |
| CongestCtrl | reach | When observing good (bad) networking conditions, the sender does not decrease (increase) packet sending rates. |
| | mono | When observing better networking conditions, the sender increases packet sending rates by either the same or a larger amount. |
| | temp | When the networking condition changes from bad to good, the sender eventually increases packet sending rates. |
| LearnedSched | reach | (1) if job A's input depends on job B's output, B is not finished, then A should not be scheduled. (2) A user cannot get their jobs scheduled earlier by requiring more resources for them. |
| AdaptBitrate | reach | When facing good (bad) downloading conditions, the video streaming system should not pick the worst (best) video resolution. |
| | mono | Better downloading conditions imply better resolutions. |

Figure 2: A high-level description of the specifications for the applications in NN4SysBench.

tasks with dependencies among them. A learned scheduler uses neural networks to learn workload-specific scheduling algorithms to optimize some high-level objective, such as minimizing average job completion time. Inputs are the status of the current jobs, tasks, and the cluster; Outputs are the next tasks to run.

**Benchmark selection.** To ensure benchmark relevance and quality, we considered the following criteria in selecting our benchmarks. First, to provide a comprehensive view of NN4Sys, we aim to cover a broad spectrum of applications in computer systems, including databases (LearnedIndex and CardEsti), networked systems (CongestCtrl and AdaptBitrate), distributed systems (LearnedSched), and generic data structures used in systems (BloomFilter). Second, we focus on highly cited works to ensure the benchmarks are impactful in their fields. Third, we prioritize the open-sourced projects, so we have accurate guidance to replicate the networks in a modern DL framework like PyTorch. Finally, we tend to choose benchmarks that have been previously subjected to verification efforts by other people [15, 50], as these NN4Sys applications are what people want to verify and we can borrow intuitions from their specification design.

## 3.2 NN4Sys specifications

Specifications are safety properties that developers expect the NN4Sys to satisfy. For example, the specification for the learned index aims to guarantee that the outputs (i.e., the predicted data positions of a key) closely align with the actual data positions, within a predefined error bound. This ensures that the learned index consistently locates existing data within the database, an essential requirement for any index structure. The correctness and safety properties of various NN4Sys applications vary significantly and are specific on the requirements of applications and the problems they solve. NN4SysBench has a diverse collection of NN4Sys, hence having a diverse set of specifications. We brief the specifications for each NN4Sys in Figure 2. In NN4SysBench, we design the specifications for each NN4Sys based on our understanding of the original systems. We also borrow specifications from prior verification work [15, 52] to some of these NN4Sys. We classify specifications to four categories: reachability, monotonicity, probabilistic, and temporal specifications.

**Reachability specification.** In some NN4Sys, developers want to bound the outputs of a neural network regarding a range of inputs, for example in learned indexes. We call these input-output properties, *reachability specifications*, which we define as: Formally, consider a neural network as a function $f$ to which inputs denote as $x \in \mathcal{D}_x$ and outputs are $y \in \mathcal{D}_y$, where $\mathcal{D}_x$ and $\mathcal{D}_y$ are the domain and range of $f$. Users can define a reachability specification by providing a pair of domains as $\langle x \in \mathcal{X}, y \in \mathcal{Y} \rangle$, where $\mathcal{X} \subseteq \mathcal{D}_x$ and $\mathcal{Y} \subseteq \mathcal{D}_y$. And a reachability specification is written as: $\forall x, \; x \in \mathcal{X} \implies y = f(x) \in \mathcal{Y}$. Reachability specification is also called *neural network robustness* in verifying vision models [40, 55, 10] and others.

**Monotonicity specification.** Monotonicity is a widely used correctness property in systems. For example, if the network condition improves, a video streaming system should not decrease the video

quality. We define monotonicity specification as: For a network $f$, two inputs $x_0$ and $x_1$ in some input domain $\mathcal{X}$, a monotonically increasing specification reads as: $\forall x_0, x_1 \in \mathcal{X}, \ \forall i, x_0[i] \geq x_1[i] \implies f(x_0)[j] \geq f(x_1)[j]$, where $j$ is an output dimension provided by users.

**Probabilistic specification.** This specification describes a NN4Sys obeying rules with some given probability. It is useful when either the guarantees hold probabilistically (e.g., in probabilistic data structures) or users want to give some leeway to NN4Sys (e.g., allowing few false predictions). For example, learned bloom filters require probabilistic specification because they allow a small number of false positives. We define probabilistic specifications as: Given a network $f$ and its input and output $x$ and $y$, users can define a probabilistic specification by specifying a reachability specification $\langle \mathcal{X}, \mathcal{Y} \rangle$ and a probability $\mathcal{P}$ indicating how likely the probabilistic specification holds. The probabilistic specification is written as: $\forall x, \ x \in \mathcal{X} \implies Pr(y \in \mathcal{Y} \mid y = f(x)) > \mathcal{P}$.

**Temporal specification.** Many NN4Sys require properties related to time, especially those applications interacting with environments. For example, a congestion control protocol should eventually increase the packet sending rate, if the network remains in good conditions. Note that the NN4Sys's decisions (here, the sending rates) may not change immediately, but the expected behavior should happen in a finite number of steps.

## 3.3  Benchmark organization

In NN4SysBench, each application (e.g., learned index) forms a *benchmark set*. Each set contains multiple *benchmark cases*. A case represents a verification problem—given a trained neural network and a set of specifications, does the network satisfy the specifications.

NN4SysBench comprises six benchmark sets, each corresponding to a NN4Sys application in Figure 1. Within each set, we've provided multiple pre-trained networks of varying sizes, reflecting a spectrum of verification challenges. Specifications are derived from our predefined templates, and users of NN4SysBench have the flexibility to configure both a random seed and the number of specifications to generate—these are two parameters in benchmark configurations. Besides network sizes, different specification templates also present varying levels of complexity. Also, as a technicality, unlike other applications, BloomFilter and LearnedSched each use a single model: BloomFilter employs a simple model, whereas LearnedSched is based on GNN, for which we retain the original model size to preserve verification complexity.

**Tuning benchmark difficulty.** NN4SysBench serves as performance tests for neural network verifiers, requiring a diverse range of difficulties to effectively tell the capabilities of the verifiers. To provide varied levels of difficulties, we alter the networks and adjust the specifications by tuning several influencing factors, including network depth, network size, the number of perturbed dimensions in specifications, and the range of specification perturbations. The interplay between these factors and the verification difficulty is studied in section 5. This investigation helps us fine-tune the benchmark difficulty to examine the verifiers' performance under varying conditions.

**Building benchmarks from scratch.** Beyond "ready-to-verify" benchmark cases, we also provide infrastructures for users who want to update the network architectures or modify specifications. Users can build their own benchmark cases with the desired neural networks and specifications. In particular, we provide the training code for each application based on PyTorch and specification templates written in VNN-LIB format [1].

## 4  Designing compatible benchmarks

As mentioned earlier (§2), some of the networks and specifications are not supported by current neural network verifiers. To make NN4SysBench accessible to and compatible with today's NN-verification, we build the *compatible NN4SysBench*. In particular, we customize the networks and specifications to satisfy the assumptions that verifiers make—the rules used by the International Verification of Neural Networks Competition [3]. Below we detail how we approximate the unsupported characteristics.

**Dual-model to simulate monotonicity.** In order to support monotonicity specification (§3.2), we implement a *dual-model* architecture that simulates two network inferences at the same time. For example, in learned cardinalities [25], we duplicate the pre-trained model and connect the two models side-by-side as a single new model (see Figure 3). The dual-model's inputs and outputs are doubled compared to the original model. By using a split operator, we divide the inputs in half: the first half

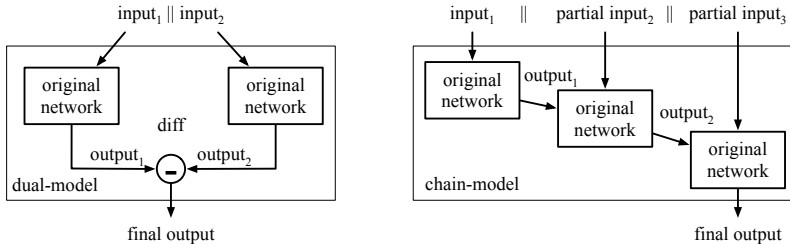

Figure 3: Dual-model and chain-model overview. In input tensors, "||" represents tensor concatenation. In the dual-model, the "diff" operator is based on the output of the original neural network. (Take CardEsti as an example: the output is the number of returned rows, so the "diff" is the subtraction operator.) In the chain-model, the "output" from the prior network and the "partial input" together form the complete input tensor for the next network in the chain.

is channeled to the first model, while the latter half is directed to the second model. Dual-model's output is the difference between the estimated cardinality for the first-half and the second-half inputs.

With the dual-model, a monotonicity specification can be expressed as, for example: (except for $year$, other properties of $q_1$ and $q_2$ are the same.) $q_1.year < 2015 < q_2.year \implies y_1 - y_2 \leq 0$, where $q_1$ and $q_2$ serve as inputs to the first and the second models respectively, while $y_1, y_2$ are the outputs from the the first and the second models, respectively.

**Chain-model to approximate $k$-step temporal specifications.** Multiple NN4Sys applications require to incorporate temporal specifications into their operations. Take, for instance, the case of Aurora [21], a system designed to tackle Internet congestion control with neural networks. In this context, users anticipate observing a gradual increase in the packet sending rate as network congestion subsides, a change that might not manifest instantaneously but should eventually happen, as long as the network conditions remain stable. These temporal specifications, which we formally defined in §3.2, play a crucial role in achieving desired system performance. However, while some verification frameworks, such as vegas [51], support verification for specific multi-step NN, verifiers still haven't fully supported temporal specifications. This highlights a significant gap in verification capabilities to ensure the safety of NN4Sys applications operating within dynamic environments.

To simulate the temporal specifications, we construct *chain-models* by replicating the original network $k$ times and concatenating them. This chain-model serves as a representation of the original model's behavior over a span of $k$ consecutive steps. Notably, the output from one model step serves as the input for the subsequent step, as illustrated in Figure 3. Consequently, for a chain-model, the cumulative inputs amount to $k$ times the inputs of the original model, while the final output corresponds to the output of the ultimate model in the sequence. This approach allows us to model the temporal specifications required for congestion control, which specify that after $k$ successive steps of favorable network conditions, the model should opt to increase the packet sending rate. Chain-model is not new; it has been used before in NN-verification by Eliyahu et al. [15] and Wu et al. [52].

**Counting units for probabilistic specifications.** Accurately verifying probabilistic specifications like learned bloom filters [39, 47] is challenging. Instead, we simulate this verification by (i) splitting the input space into tiny equal-sized units, (ii) verifying each unit separately, and (iii) counting the verified safe units over all units as the probability of verified safe space. Of course, this is a conservative approximation because a verified unsafe unit might have most of the space that is safe. The size of the unit will be a parameter to trade off verification performance and the accuracy of probabilistic specifications. For our learned bloom filter benchmark, we choose a dataset that has a two dimensional input (a geolocation with a latitude and a longitude). We split the input space evenly to 10K (i.e., $100 \times 100$) units.

**Merging hierarchical models for learned index.** It is hard to verify vanilla RMIs [26] as a whole because an RMI has a hierarchical structure and is comprised of many models. In NN4SysBench, we use a straightforward approach to approximate learned index: employing a single neural network to learn what an RMI learns. However, neural networks are more costly and harder to train compared to RMIs, as noted by Kraska et al. [26, §2.3]. To address this, we borrow training approaches from Ouroboros [50] to train a single neural network that learns very well in a 150K-key lognormal dataset.

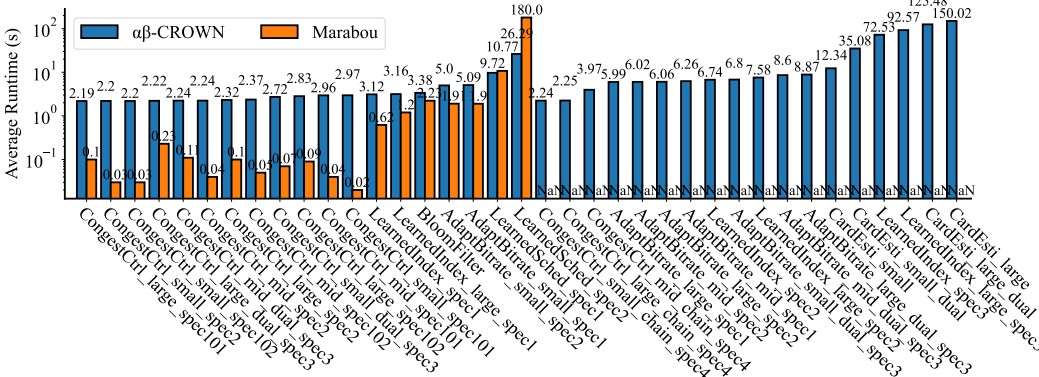

Figure 4: Verification runtime for different benchmarks. The x-axis are the benchmarks; the y-axis is the average verification runtime of ten cases (in log-scale). Bars on the left are the benchmarks solved by both verifiers; on the right are the ones only solved by $\alpha\beta$-CROWN. The bars are sorted by verification time, with the exact runtime displayed above each bar. Instances labeled "NaN" indicate cases where the given verifier cannot verify the instance.

## 5 Experimental evaluation

There are two questions we want to answer:

- How do state-of-the-art verifiers perform on NN4SysBench?
- How does verification difficulty change according to different benchmarks?

**Verifiers.** We experiment with two verifiers: (1) $\alpha\beta$-CROWN [4] is a verifier based on an efficient linear bound propagation framework and branch and bound. It can be accelerated efficiently on GPUs and can scale to large networks. (2) Marabou [5] is an SMT-based verifier that transforms neural network verification to constraint satisfaction problems, and solves the problem using a specialized SMT solver (github commit `a2077b46`). Note that the verifier performance comparison is not entirely apple-to-apple, as $\alpha\beta$-CROWN uses GPUs and Marabou does not.

**Benchmarks.** Each benchmark comprises (1) a trained neural network and (2) a set of specifications, named after their combination. For example, `CongestCtrl_small_chain_spec4` represents a benchmark from application CongestCtrl: the model size is small; it is a chain-model (§4); the corresponding specification is `spec4`. We briefly describe specifications: (1) LearnedIndex: all specifications are reachability specifications, different in the number of spec entries. (2) CardEsti: specifications for dual-model are monotonicity specifications, for normal models are reachability specifications. (3) BloomFilter: specifications are approximated probabilistic specifications (§4) (4) CongestCtrl: `spec101/102/2` are reachability specifications; `spec3` is monotonicity specifications for dual-models; `spec4` is temporal specifications for chain-models. (5) LearnedSched: specifications (`spec1/2`) are reachability specifications. (6) AdaptBitrate: specifications (`spec1/2`) are reachability specifications; `spec3` is monotonicity specifications for dual-models. Please see Figure 2 for the meaning of each type of specifications regarding different NN4Sys.

**NN4SysBench configs.** NN4SysBench allows users to configure the number of specifications for each benchmark, and randomly generate specifications (§3.3). In the following experiments, we fix the random seed to generate the same set of benchmarks for two verifiers, and we configure NN4SysBench to generate 10 specifications from one specification template. That means for each pair of application and specification template, we generate 10 cases.

**Experiment setup.** We run experiments on a machine with 32-Core Intel Xeon Gold 6338 CPU, 256GB memory, and an NVIDIA A30(24GB) GPU. Detailed settings can be found in Appendix.

### 5.1 NN4Sys verification performance

We experiment all the above mentioned benchmarks on the two verifiers, $\alpha\beta$-CROWN and Marabou. We measure how much time each verifier spends on the verification. In this experiment, we collect the end-to-end verification time for each benchmark. Note that we configure NN4SysBench to generate

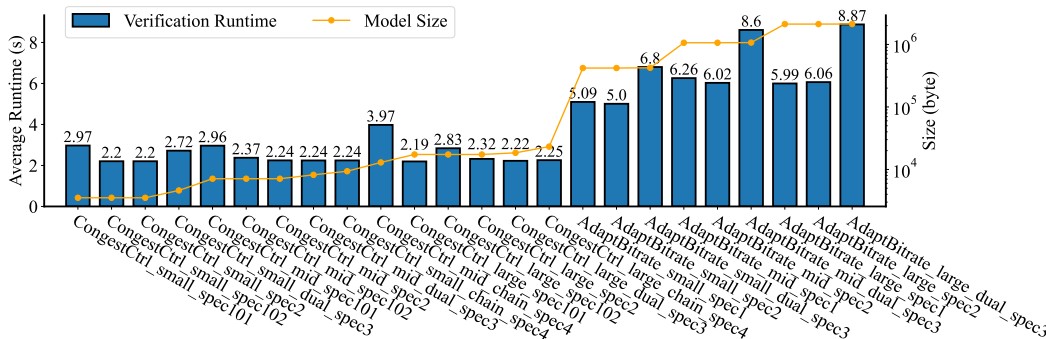

Figure 5: Verification time versus model size for CongestCtrl and AdaptBitrate. The bars represent the benchmark verification time from $\alpha\beta$-CROWN (the left y-axis), and the line indicates the `ONNX` model size in bytes (the right y-axis). The benchmarks are sorted by their model size.

10 cases for each benchmark, so the runtime for each benchmark is the average of ten runs. If there is a timeout (180 seconds), we add 180 seconds to the overall time. Figure 4 shows the results.

As Figure 4 shows, different benchmarks have significantly different verification runtime (notice that y-axis is in log-scale). This highlights that NN4SysBench can distinguish verification capabilities. Also, all benchmarks can be verified by at least one of the verifier, indicating that our compatible benchmarks indeed work with today's verifiers. Finally, we can see that different verifiers perform differently on different benchmarks. For example, Marabou can solve some small networks very efficiently; meanwhile, $\alpha\beta$-CROWN can solve some large networks that Marabou cannot address.

Even in compatible version of NN4SysBench, there are several benchmarks that neither verifier is able to verify, due to unsupported operators. In particular, the $\alpha\beta$-CROWN does not support scatter operator, which are used in chain-models, such as `LearnedSched_chain` and `AdaptBitrate_chain` models. Meanwhile, Marabou does not support `Add(const)`, `Matmul(var)`, slice operator, and some other operators, making it unable to verify a number of benchmarks.

## 5.2 Benchmark difficulty

NN4SysBench is designed to explore different verification difficulties. This demonstrates the intrinsic complexity of each NN4Sys application. In addition, various difficulties allow NN4SysBench, as a benchmark suite, provide more information for different verifiers. Next, we study what's the difficulties of different NN4Sys applications, and different specifications. We use the verification time of $\alpha\beta$-CROWN as a metric to tell the benchmark difficulty. We experiment with two NN4Sys applications (CongestCtrl and AdaptBitrate) with multiple specification templates. We study their difficulties regarding their verification time and model sizes. Figure 5 shows the results.

There are three trends in Figure 5. First, the overall trend is that the larger the model, the longer the verification time. Hence, the verification difficulty increases when the model size increases. This is not surprising, as the number of non-linear operations decides the potential branches during verification. Second, for these two applications, monotonicity specifications are harder to verify than reachability specifications. For example, verifying `AdaptBitrate_small_dual_spec3` takes longer than verifying `AdaptBitrate_small_spec2`, even though they have similar model sizes. Third, safe specifications are harder to verify than unsafe specifications. For example, given same model, verifying `CongestCtrl_small_spec101`, which contains 10 safe cases and 0 unsafe cases, is longer than verifying `CongestCtrl_small_spec102`, which contains 0 safe and 10 unsafe cases.

# 6 Related work

**Benchmarks in verification.** Benchmarking in verification has developed in response to the empirical research within different fields [18, 14]. Compared to these developed fields, neural network verification is a new field under development, and verification competitions have been positive driven force to develop high-quality benchmarks. In particular, International Verification of Neural Networks Competition [8], VNN-COMP, has been a representative competition to introduce and compare state-of-the-art methods in NN-verification. It also has benchmarks spanning different applications and scenarios.

Carvana UNet [2] is a benchmark proposed in VNN-COMP that introduces networks and specifications for semantic segmentation in autonomous driving. To benchmark the ability that current verifiers handle practical neural network architectures, Carvana UNet covers comparably complex networks including Conv2d layers, AveragePool layers and TransposedConv Upsampling [46] layers followed by batch normalization. ERAN benchmark is proposed to understand how the choice of activation function affects certifiability. Although most of NN Verification methods focus their analysis on ReLU based networks, modern network architectures, such as EfficientNet [44], are based on non-piecewise-linear activation functions. The ERAN benchmark aims at comparing the certifiability of networks based on piecewise-linear and non-piecewise-linear activation functions. Xu et al. [53] summarizes a benchmark suite on various tasks with networks, most of which are composed of fully-connected layers.

NN4SysBench is different from these literatures as it focuses on verification for neural networks within system applications, advocating to bridge the gap between current NN-verification methods and growing machine learning for system applications. The proposed benchmark provides supplementary tasks, architectures, and specifications to the verification community, and shows the potential to advance research in NN-verification for systems.

## 7 Future work and conclusion

**Future work.** We're working on adding more NN4Sys instances to NN4SysBench, for example, I/O latency predictors [20], OS kernel load balancing [16], and learned memory allocator [31]. One interesting topic is to study and provide a set of high-level specification interfaces for NN4Sys that cover most safety properties required by computer systems. We also plan to include other categories of specifications, and develop sophisticated specifications that have more structures than a simple parallel OR in VNN-LIB. In particular, we plan to build compound specifications which are a combination of multiple different categories of specifications to serve the same safety property. For example, a compound of monotonicity and error-bounded existing keys can substitute the current specifications for learned index, with much fewer specification entries.

**Conclusion.** We present a benchmark suite, NN4SysBench, with a hope to bridge NN-verification and NN4Sys. NN4SysBench is designed for today's verification tools while hinting at the future verification of NN4Sys.

## Acknowledgements

This work has been partially supported by Khoury apprenticeship program. Cheng Tan is supported in part by NSF CAREER #2237295. Huan Zhang is supported in part by the AI2050 program at Schmidt Sciences (AI 2050 Early Career Fellowship) and NSF (IIS-2331967).

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
