# A  Hosting, licensing, and maintenance

**Github** Our benchmark, code and document are available here: `https://github.com/Khoury-srg/NN4SysBench`.

**Licensing** Our benchmark and code are available under a MIT license.

**Maintenance** We're working on adding more NN4Sys instances to NN4SysBench.

# B  Details of training

In our experiments, we followed the same training steps and used similar settings as described in the original NN4Sys application paper, which allowed us to compare our results with theirs. Originally, the models were implemented in TensorFlow, but we re-implemented them in PyTorch. Due to this change in the library, there might be slight differences in the training outcomes and model performance. For verification purposes, we aimed for our models to achieve at least 80% of the performance reported in the original paper. Some important details are explained as follows.

**Learned adaptive bitrate** In the original paper, three types of rewards were offered for training the model. We chose to use the linear reward as it produced the best performance for the model.

# C  Details of generation

First, we create thousands of instances for each specification to form an input pool. These instances follow our intuitive specification guidelines. When creating benchmarks, we randomly select 10 instances for each specification, based on the random seed. Additionally, we introduce a range to the input to perturb this dimension. Some important hyperparameters are explained as follows.

## C.1  Which dimension to perturb

We perturb the dimensions that are related to the definition of the specification.

- **Database learned index** We perturb the key.
- **Learned bloom filter** We perturb the data item.
- **Learned cardinalities** We perturb the query.
- **Learned Internet congestion control** We perturb latency gradient and latency ratio for specification 1.1, 3 and 4. We perturb latency gradient, latency ratio and sending ratio for specification 1.2 and 2.
- **Learned adaptive bitrate** We perturb chunk throuput and download time.
- **Learned distributed system scheduler** We perturb the average task duration.

## C.2  Range

We chosen the range that satisfies two conditions: first, the verification time stays within a time limit (approximately 5 seconds for simple specifications, 10 seconds for medium specifications, and between 30 to 60 seconds for more challenging specifications); second, the inclusion of the range still adheres to the definition of intuitive specifications.

# D  Experiment setup

We run experiments on a machine with 32-Core Intel Xeon Gold 6338 CPU, 256GB memory, and an NVIDIA A30(24GB) GPU. The OS is Ubuntu 22.04. Verifier Marabou runs with Python 3.10.9, PyTorch 1.12.1, yaml 0.2.5, numpy 1.23.5, and onnx 1.13.1; $\alpha\beta$-CROWN runs with Python 3.9.17,

39 PyTorch 1.11.0, yaml 0.2.5, numpy 1.21.0, onnx 1.13.1, and onnx2pytorch 0.4.1. We run the two
40 verifiers with different software versions because the verifiers require so.