# OpenReview forum: "NN4SysBench: Characterizing Neural Network Verification for Computer Systems"
_NeurIPS.cc/2024/Datasets_and_Benchmarks_Track — NeurIPS 2024 Track Datasets and Benchmarks Poster_

### Official Review · Reviewer_15jd · 2024-07-22
**Good contribution that presents new challenges in neural network verification**

**Rating:** 7
**Confidence:** 4

**Review:**

Overall, the benchmark provides a significant contribution that will promote new research in the area of verifying NN4Sys.

Pros:
- The benchmark examples were selected carefully and are well-motivated (last paragraph of section 3.1).
- The compatible version makes the benchmark accessible to current neural network verification researchers (even if they do not have specific knowledge of computer systems).
- The vanilla version presents new challenges that encourage further advancements in neural network verification.
- The ability to tune the difficulty of the verification problem is a nice feature

Cons:
- The paper could benefit from some further explanation and improved grammar is some areas.

**Strengths:**

Same as pros in the review above:

- The benchmark examples were selected carefully and are well-motivated (last paragraph of section 3.1).
- The compatible version makes the benchmark accessible to current neural network verification researchers (even if they do not have specific knowledge of computer systems).
- The vanilla version presents new challenges that encourage further advancements in neural network verification.
- The ability to tune the difficulty of the verification problem is a nice feature

**Additional Feedback:**

Questions:

- For the learned indexing example, if I am understanding correctly, the risk is that the neural network could output a data position for a nonexistent key. Could this just be checked online when running the system without the need to formally verify ahead of time?
- For the dual-model that simulates monotonicity, it would be helpful to provide some more explanation on the approximation that is being introduced. For the year example, would you have to check this specification for all possible years to prove monotonicity?

**Clarity:**

The paper is logically organized and easy to follow. However, it could benefit from some proofreading to fix grammatical errors.

**Correctness:**

To the best of my knowledge, the claims are correct. However, there is one area that could use some additional support. Specifically, in section 5.2, why not also use Marabou to assess benchmark difficulty? How do we know that the trends are not specific to $\alpha\beta$-CROWN?

**Documentation:**

The main README contains information on how to reproduce the experiments. However, the repository could benefit from additional comments and in the code and an explanation of what each file does/which files are most important.

**Limitations:**

Adequately addressed (lines 117-124)

**Opportunities For Improvement:**

- Line 98: properties -> property
- Line 104: What is a specification entry?
- Line 188: this -> these
- Check grammar in Reachability specification paragraph (some sentence fragments)
- Need explanation for NaNs in Figure 4
- Line 333: provide -> to provide
- Lines 258-260: There is a lot of work in verifying neural feedback loops that is probably at least worth mentioning here

**Relation To Prior Work:**

The authors adequately discuss what makes verifying neural networks for computer systems different from other verification problems and provide context in previous literature.

**Summary And Contributions:**

This paper presents a new set of neural network verification benchmarks that relate to neural networks used in computer systems, which present additional challenges beyond the existing systems neural network verification tools have been designed for. The authors select a set of benchmark problems from a variety of applications that highlight the various challenges. Because existing neural network verification tools are not yet compatible with some of these systems, the authors provide both a vanilla version of the benchmarks and a compatible version that can be verified directly with existing tools.

---

> ### Author Rebuttal · Authors · 2024-08-19
>
> **Improving grammar and fixing typos**
>
> Thank you for the comments! We will fix the identified typos and conduct a
> thorough proofreading for the revised version of this paper.
>
> **What is a specification entry?**
>
> A "specification entry" is the basic unit of a specification, representing a
> single rule. For example, in reachability specifications, an entry consists of
> an input range and an output range.
>
> We will clarify this in our paper.
>
> **Need an explanation for NaNs in Figure 4.**
>
> Thank you. "NaN" means this instance cannot be verified by
> this tool. We will make it clear in the paper.
>
> **In S5.2, why not use Marabou to assess benchmark
> difficulty? How do we know that the trends are not specific to ab-CROWN?**
>
> The trend also applies to Marabou. Please see the pdf attached. We didn't plot
> Marabou because there are specifications that Marabou cannot verify (marked as
> NaN).
>
>
> **The main README lacks information**
>
> Thank you for the feedback! We will improve our documentation as mentioned in
> the general questions.
>
>
> **For the learned index, could this just be checked online when running the system
> without the need to formally verify ahead of time?**
>
> Yes, online checking can correct the output. However, in the worst case (i.e., the
> output is very wrong), it requires loading many blocks from the disk, which is
> expensive. Verification, on the other hand, provides a lower bound for this
> worst-case scenario.
> We will clarify this in the paper.
>
> **The dual-model: for the year example, would you need to check this
> specification for every possible year to prove monotonicity?**
>
> For full monotonicity, yes, the current verifiers require checking all possible
> years. This is a caveat of using approximate specifications in the compatible
> benchmarks. In NN4SysBench, we randomly generate a subset of years for
> verification.
>
> We will clarify this in the dual-model section of our paper.

---

### Official Review · Reviewer_1WSJ · 2024-07-25
**A Benchmark suite for NN4Sys (Neural Network for Computer Systems)**

**Rating:** 7
**Confidence:** 3
**Correctness:** Yes
**Clarity:** Yes

**Review:**

1. A comprehensive benchmark suite for NN4Sys applications with proposed safety specifications.
2. The authors have also provided the modified versions of NN4Sys applications to enable current verifiers to experiment with them apart from the original version and specifications.
3. The motivation for proposing NN4SysBench is very well-detailed in the paper.
4. Since the authors have considered several applications, their proposed safety specifications seem limited and might not apply to all NN4Sys applications in every scenario.

**Strengths:**

1. NN4SysBench covers a wide range of applications and verification challenges.
2. I liked that the authors focus on real-world applications in computer systems, such as congestion control and adaptive streaming, which have been key research areas for a very long time. hence, applying neural networks in such applications is on the boom.

**Additional Feedback:**

No

**Documentation:**

Yes

**Ethics:**

Yes

**Limitations:**

1. The authors have not highlighted the limitations of this Benchmark suite.
2. The authors have not highlighted how to replicate and utilize the proposed benchmark suite for other applications. For example, they have used the Pensieve ABR algorithm for adaptive bitrate; how can it be used on other  ABR algorithms such as Oboe, FastMPC, etc? And similarly for other congestion control algorithms.

**Opportunities For Improvement:**

1. Some more applications of computer systems can be explored as a future research to be a part of NN4SysBench which is missing in the paper as future work.
2. The limitations of the work are not detailed anywhere in the paper.

**Relation To Prior Work:**

Yes, the related work is discussed.

**Summary And Contributions:**

The authors introduced a benchmark suite named NN4SysBench, which is designed to assess the verification of neural networks used in computer systems. It is referred to as NN4Sys. The benchmark suite aims to bridge the gap between NN4Sys and neural network verification by highlighting unique challenges specific to computer systems often overlooked by current verification tools. The suite encompasses a variety of applications to showcase different verification difficulties.

---

> ### Author Rebuttal · Authors · 2024-08-19
>
> **More applications as future work?**
>
> Thank you for the suggestion! We will have a future work section
> with a plan to include more NN4Sys applications.
>
> **What are the limitations?**
>
> Thank you for the question! The major limitations are:
> - The specification categories are imbalanced: there are more reachability and
>   monotonicity specifications than probabilistic and temporal ones.
> - The diversity within each specification category and across applications is limited.
>
> We plan to keep updating NN4SysBench and develop more specifications.
> We will add the mentioned limitations to our paper.
>
> **The authors have not highlighted how to replicate and utilize the proposed
> benchmark suite for other applications.**
>
> Thank you for the feedback! We will improve our documentation as mentioned in
> the general questions.

---

### Official Review · Reviewer_7er3 · 2024-07-29
**A nice benchmark suite with less meaningful specfications**

**Rating:** 5
**Confidence:** 5
**Clarity:** The paper is well-motivated and well-…

**Review:**

I have mixed feelings about this work. On the one hand, it attempts to propose important and exciting challenges in neural network verification, which has not yet attracted sufficient attention from the community. It is certainly a nice contribution. However, on the other hand, the completion is barely minimal. It is great to point out that specifications are in four broad categories, but the prob category consists of only a single instance. The takeaway message from the evaluation is not very interesting -- larger model takes longer to be verified, and all instances can already be verified with the state-of-the-art engine.  For formal verification tasks, the quality of specifications is essential. This work acknowledges its importance but does not make much effort to get high-quality specifications, which is a major weakness, especially for a dataset and benchmark work.

**Strengths:**

- This work concerns an important verification problem of neural networks, beyond the classic computer vision applications.
- This work systematically classifies specifications into four categories.
- Benchmarks with varying sizes are provided in most applications (except for LearnedSched).

**Additional Feedback:**

Why is there only one size for LearnedSched?

Why is the size 'medium' missing in some applications?

What temporal specifications are used in the CongetCtrl?

**Correctness:**

The main concern is that specifications are generated according to some template.

**Documentation:**

The documentation is somewhat minimal, which could be greatly improved.
For instance, `run x.py` is certainly better than nothing, but is not very helpful (e.g., what is a meaningful run? how to interpret the results?) https://github.com/lydialin1212/NN4Sys_Benchmark/tree/main/Models/Bloom_filter

**Ethics:**

There are no ethical concerns.

**Limitations:**

There are no negative societal impacts of this work.

**Opportunities For Improvement:**

Meaningful specifications are essential for formal verification tasks since verification is only as good as what specifications say. To set up a strong and compelling verification benchmark, collecting a set of realistic and representative specifications in the first place is highly recommended.

**Relation To Prior Work:**

Yes, the authors made appropriate discussions regarding the specific contributions in this work.

**Summary And Contributions:**

This paper concerns neural network verifications of applications for computer systems (NN4Sys) and collects neural network instances from six applications. The corresponding specifications are in four categories: reachability, monotonicity, probability, and temporal. Evaluations with _generated specifications_ show that, when model size increases, the state-of-the-art verification engine takes longer to verify.

---

> ### Author Rebuttal · Authors · 2024-08-19
>
> **The takeaway messages from the evaluation seem trivial.
> Are there any interesting takeaway messages?**
>
> We do observe some counterintuitive results, which we will elaborate in our
> paper. For example, we initially assumed that mid-sized models, with twice the
> number of neurons per layer compared to small models, would take significantly
> longer to verify. Surprisingly, the verification time was nearly the same for
> both model sizes. Additionally, we found that monotonicity specifications are
> more challenging to verify than reachability specifications, and safe
> specifications are harder to verify than unsafe ones. We will discuss these
> findings in greater detail in Section 5.
>
>
>
>
> **Isn't that all instances already be verified by today's verifiers?**
>
> - For the _compatible benchmarks_, yes.
>   These are specifically designed to be verified by today's verifiers. In fact,
>   we had to adjust both the model architectures and specifications to enable
>   successful verification. Please see S2 and S4 for more details.
>
> - For the _vanilla benchmarks_, there are operators and specifications that
>   none of today's verifiers support. For example, NN4SysBench ~~includes~~ used to have [updated: 08/21] temporal
>   specifications for AdaptBitrate, where the specifications are written as:
>   "When in good/bad condition, the resolution should eventually
>   increase/decrease (after five steps)". However, tools like ab-CROWN and Marabou
>   could not verify these instances. ab-CROWN doesn't support index
>   ~~perturbation~~ scatter [updated:08/21], and at the time, Marabou didn't support constant addition
>   (though this has been fixed after we reported it)
>
> As verifiers evolve, we aim to reintroduce these specifications into our
> benchmark soon. We also discovered valuable insights from these unverifiable
> instances, which we shared with verifier developers, leading to improvements
> (like the above Marabou case). We plan to include these findings and additional
> instances in the revised version.
>
>
> **lacking high-quality specifications?**
>
> Designing effective specifications is challenging. We have invested substantial
> time in developing meaningful specifications for each application. To show
> this, we plan to include a discussion on the specifications, possibly
> highlighting some failed attempts to illustrate why the proposed specifications
> were chosen. We will also include in the revised version that creating
> specifications remains an open question.
>
> In addition, we will include a section on essential specifications for NN4Sys
> that are currently unsupported, with the aim of motivating the development of
> more expressive verifiers.
>
>
> **Improvement: The documentation is somewhat minimal.**
>
> Thank you for the feedback! We will improve our documentation as mentioned in
> the general questions.
>
>
> **Why is there only one size for LearnedSched?**
>
> LearnedSched is a GNN model. We keep the original size to preserve its
> verification difficulty. While adjusting the model size is feasible, it is
> hard to assess whether a resized model would still maintain the intended
> performance, given the scheduling application is hard to evaluate.
> We will clarify this in the paper.
>
>
> **Why is the size 'medium' missing in some applications?**
>
> This is a design choice: we created three model sizes for complex
> applications and two for simpler ones.
>
>
> **What temporal specifications are used in the CongetCtrl?**
>
> Intuitively, the temporal specifications read as, "When the networking
> condition changes from bad to good, the sender eventually increases packet
> sending rates." (Figure 2)

---

### Author Rebuttal · Authors · 2024-08-19

We thank the reviewers for their thoughtful comments.
We answer a general question below and respond to specific reviewer questions
under their respective reviews.

**Immature documentation.**
Reviewer 7er3 described the repo documentation as "minimal". Reviewer 1WSJ expressed
concerns about replicating and applying the benchmarks to other applications.
Reviewer 15jd noted a lack of comments and explanations in the code.

Thank you! Indeed, the current NN4SysBench repository is incomplete and
temporary. We are actively working on comprehensive documentation, which will
include detailed instructions for reproducing our experiments, more documents
for explaining the code, and how to extend our benchmarks. We expect to have
fully developed documentation available soon.

---

### Comment · Area_Chair_7Dm4 · 2024-08-31
**Feedback on rebuttals**

Dear Reviewers, Thank you for your reviews. Please review the author rebuttals and give feedback as the deadline is tomorrow for the interactions. Thank you. Best, AC

---

### Decision · Program_Chairs · 2024-09-26

**Decision:**

Accept (Poster)

**Comment:**

Since both AC and SAC are non-responsive, PCs decide to give accept based on current reviewers' comments.